# Comparison of Joint Effect of Acute and Chronic Toxicity for Combined Assessment of Heavy Metals on *Photobacterium* sp.NAA-MIE

**DOI:** 10.3390/ijerph18126644

**Published:** 2021-06-21

**Authors:** Nur Adila Adnan, Mohd Izuan Effendi Halmi, Siti Salwa Abd Gani, Uswatun Hasanah Zaidan, Mohd Yunus Abd Shukor

**Affiliations:** 1Department of Land Management, Faculty of Agriculture, University Putra Malaysia, Serdang 43400, Selangor, Malaysia; nuradilaadnan09@gmail.com; 2Department of Agricultural Technology, Faculty of Agriculture, University Putra Malaysia, Serdang 43400, Selangor, Malaysia; ssalwaag@upm.edu.my; 3Department of Biochemistry, Faculty of Biotechnology and Biomolecular Sciences, University Putra Malaysia, Serdang 43400, Selangor, Malaysia; uswatun@upm.edu.my (U.H.Z.); mohdyunus@upm.edu.my (M.Y.A.S.)

**Keywords:** toxicity assessment, luminescent bacteria, acute and chronic toxicity

## Abstract

Predicting the crucial effect of single metal pollutants against the aquatic ecosystem has been highly debatable for decades. However, dealing with complex metal mixtures management in toxicological studies creates a challenge, as heavy metals may evoke greater toxicity on interactions with other constituents rather than individually low acting concentrations. Moreover, the toxicity mechanisms are different between short term and long term exposure of the metal toxicant. In this study, acute and chronic toxicity based on luminescence inhibition assay using newly isolated *Photobacterium* sp.NAA-MIE as the indicator are presented. *Photobacterium* sp.NAA-MIE was exposed to the mixture at a predetermined ratio of 1:1. TU (Toxicity Unit) and MTI (Mixture Toxic Index) approach presented the mixture toxicity of Hg^2+^ + Ag^+^, Hg^2+^ + Cu^2+^, Ag^+^ + Cu^2+^, Hg^2+^ + Ag^+^ + Cu^2+^, and Cd^2+^ + Cu^2+^ showed antagonistic effect over acute and chronic test. Binary mixture of Cu^2+^ + Zn^2+^ was observed to show additive effect at acute test and antagonistic effect at chronic test while mixture of Ni^2+^ + Zn^2+^ showing antagonistic effect during acute test and synergistic effect during chronic test. Thus, the strain is suitable and their use as bioassay to predict the risk assessment of heavy metal under acute toxicity without abandoning the advantage of chronic toxicity extrapolation.

## 1. Introduction

The victory of urbanization, the agricultural and pharmaceutical industries, and other economic developments make important contributions to our health and high living standard. Nevertheless, their application is often correlated with persistent problems of heavy metals in solid and liquid wastes and contributes to the pollution of the environment [1,2]. Bioaccumulation properties of metals can drive high biotic tissue concentrations in consumers, predominantly through food webs even less than the range of acute toxic concentrations [3,4]. Globally, improper management of municipal solid waste incineration and sewage is giving rise to mercury (Hg) increasing from 0.5 to 9.0 µg/L in the aquatic environment of China [5,6,7]. In other cases, Cu and As contaminated areas of the Rhône River (France) are constantly increasing due to mining activities and pesticides in organic agriculture [8,9]. In Malaysia itself, as an important international port, in three subsidiary ports in Port Klang, concentrations of metals As, Cd and Pb were comparatively higher than the background values in the coastal sediment [10].

Unfortunately, the analysis of the total amount of heavy metals is unable to represent the full environmental behaviors and ecological effects of heavy metals [11]. Numerous studies deal with risk effects of single metal types, while in reality the aquatic organisms are often exposed to mixtures of metals [12,13]. Thus, the effects of co-existing multiple chemicals cannot be ignored and interactions of components in a mixture might cause significant changes in the properties of its constituents [14]. Indeed, organisms are susceptible to multiple mixtures of metals in ecosystems, which may have antagonistic, synergistic or additive effects, and complex formations may critically lead to an imbalanced ecosystem [15,16]. In the event two or more chemicals are found concurrently in organisms, the joint effect may bring in the addition of the toxic effect of one chemical to the other, known as additive interaction. Furthermore, antagonistic interaction occurs if the toxic effects created by the mixture are lower than the sum of the toxic effects of the individual elements, and vice versa, synergistic interaction takes place if toxic effects of metal present in the mixture go beyond the total effects of the individual components [17,18].

In current regulatory approaches, bioassays evaluate the physiological or behavioral changes exhibited by living organisms attributed to metabolic disruption caused by toxic compounds [19,20]. Existing standard toxicity tests both on prokaryotic and eukaryotic species, such as algae, fish and daphnid, are expensive; these tests give slow responses and entail a long life cycle to reproducibility [21,22]. Taken for example, Nwanyanwu and co-workers studied the toxicity of a binary mixture of Zn and Cd based on the inhibition of dehydrogenase activity of three bacteria consortium. However, the time taken for the toxicant to react with these bacteria was 24 h [15].

In this regard, it is mandatory to determine the interactive effects of the mixtures of pollutant on environmental microbial associated mixtures of pollutants with the demand of an ecotoxicity test that is cost-effective, rapid response (5 min–30 min), sensitive and reliable. Bioluminescence inhibition assay promises further advantage of easy use, low cost and high reproducibility based on reduction of the light output emitted by naturally luminous bacteria such as *Vibrio fischeri* and *Photobacterium phosphoreum* in the presence of toxicants due to the inhibition of enzyme luciferase [19,23,24]. This bioassay has been accepted as a quick method for chemical toxicity assessment and commercialized as Microtox^®^ (Azur Environmental, California, USA), BioTox™ (Aboatox, Masku, Finland), ToxAlert^®^ (Merck, Darmstadt, Germany) test for ecotoxicological monitoring and chemical testing of wastewater effluents, sediment extracts and contaminated groundwater [19,25]. Even so, this system needs to maintain optimum assay temperatures, which range from 15 to 25 °C. Therefore, the system is not practical in order to conduct an assay in a tropical country like Malaysia, which displays a broad variation of regular temperatures up to 34 °C in the mid-day [26]. A slight change out of this range of temperature could extremely affect the luminescence activity [24]. Besides, the system would depend on a refrigerated water bath to operate, which is not constructive, expensive and difficult to maintain, and instrument-dependent for field applications in the tropical regions [27]. Hence, a cost-effective bioluminescence inhibition assay using local isolate without the application of expensive luminometer and instruments, and, a higher sensitivity towards heavy metals is urgently needed in tropical regions.

At the moment, bioluminescence inhibition assay has been widely used to test acute toxicities of metals. Based on previous studies, *Vibrio fischeri* bioluminescence inhibition bioassay (VFBIA) only provides information about the overall acute metal toxicity [11,17,19]. Direct assessment of the potential chronic effects of metals by applying naturally bioluminescence species is currently lacking. Blasco and Del [28] and Wang et al., [29] recommended that chronic tests could be more applicable to signify a threat to public health. It can be argued that feedback at the chronic joint may be more crucial, because of the occurrence of compounds in the environment for an extended period of time. Thus, it is necessary to study these chronic joint effects. In the prior work, luminescence activity of luminescent bacterium identified as *Photobacterium* sp.NAA-MIE isolated from local marine fish *Selar crumenophthalmus* has exhibited temperature and pH stable in the tropical environment and sensitive towards heavy metals [30]. In this study, acute and chronic metal toxicity for mixture of heavy metals were conducted using this strain without the application of expensive instrumentation. The toxicity results of *Photobacterium* sp.NAA-MIE will be helpful in the future for assessing the individual and joint toxicity during the treatment of industrial wastewater.

## 2. Materials and Methods 

The steps of the experimental procedure are summarized in Figure 1.

### 2.1. Preparation of Bacterial Culture and Media for Toxicity Assay

Luminescent bacterium *Photobacterium* sp.NAA-MIE newly isolated from local marine fish *Selar crumenophthalmus* [28] was used in this study. 1.0% (*v*/*v*) of bacterial culture (OD_600_ = 0.7–0.8) was inoculated into luminescence broth medium by dissolving 10 g of NaCl, 10 g of peptone, 3 mL of glycerol and 3 g of yeast in 1 L of distilled water [20] and grown at room temperature for 12 h on rotary shaker (100 rpm).The culture was then harvested by centrifugation (10,000× *g*, 10 min) and the pellet was diluted using 1 L of minimal salt media (MSM) (12.8 g of Na_2_HPO_4_ 7H_2_O, 3.1 g of KH_2_PO_4_, 17 g of NaCl, 1 g NH_4_Cl, 0.5 g MgSO_4_, and 3 mL glycerol) as substitute media for toxicity assay. The luminescence and sensitivity bioassay can be limited by a high density of the bacterial culture. To overcome this, the culture was further diluted 1000 times until the emission was within the range of 40,000 to 90,000 RLU before assaying [22,31,32].

### 2.2. Preparation of Heavy Metal Solutions

For individual metal toxicity test seven heavy metals were used: Hg^2+^, Cu^2+^, Ag^+^, Pb^2+^, Cd^2+^, Ni^2+^ and Zn^2+^. Standard procedure was adopted to prepare metal stock solutions ranging over (0.5, 1, 10, 100 and 1000 ppm or mg/L) by dissolving the AAS grade metal in deionized water. The solutions were freshly prepared [13]. Ten concentrations of the tested metals were prepared from stock solutions to carry out a preliminary experiment in triplicate to determine the suitable metal concentration array for determining the half maximal inhibitory concentration IC_50_ of each tested metal. All analytical-grade chemicals used were purchased from Oxoid (North Shore, England), and Merck (Damstardt, Germany).

### 2.3. Acute and Chronic Toxicity Assay of Single Metals

The assay was conducted by mixing 10 µL of toxicant solutions with 190 µL of bacterial cultures. Deionized water was used as a control to replace the toxicants in this study. The IC_50_ values were determined at total time per assay, approximately 30 min for acute toxicity and 6 h incubation for chronic toxicity, at room temperature (26 °C). The luminescence was determined with a portable luminometer (Lumitester PD-30 by Kikkoman, Tokyo, Japan) and reported as Relative Luminescent Unit (RLU). A total of 200 µL of bacterial culture was inoculated into the LuciPac pen tube before the readings were taken. Each assay was performed at least in triplicate.

### 2.4. Acute and Chronic Toxicity Assay of Metal Mixture

Concentration for each selected metal mixture was prepared in an equitoxic ratio of 1:1 and 1:1:1 based on concentrations of each toxicant producing 50% light reduction when being tested individually [33]. In this study, Cu + Cd, Cu + Zn, Ni + Zn, Hg + Cu, Hg + Ag, Ag + Cu and Hg + Ag + Cu mixtures were proposed to evaluate possible effects for each combination. The tests for mixtures were conducted similarly to those for the individual metals for both acute and chronic toxicity at 30 min and 6 h exposure time.

### 2.5. Data Analysis

#### 2.5.1. Half Maximal Inhibitory Concentration (IC_50_) Study of Metals Toxicant on Luminescence Production

The bioluminescence inhibition study allows identifying the concentration (IC_50_) of the toxicant (mg/L) that gives a 50% reduction in light [22,34]. The luminescence activity of bacteria samples was inhibited by a series of the studied metals at different ranges of concentrations (mg/L) after exposure for 30 min (acute) and 6 h (chronic). The variation in light reduction and concentration of the toxicant produces a response relationship. IC_50_ calculations according to previous works was outlined by [35,36]. The percentage of luminescence inhibition was determined and IC_50_ was a nonlinear regression One Phase Decay model for one-phase binding and four-parameter logistic models software calculated using Graphpad Prism version 7.04 (California, USA). Means and standard errors were determined according to at least three independent replicates of each test.

#### 2.5.2. Measurement of the Luminescence Inhibition

According to [37] intensity of luminescence produced by luminescence bacteria is inhibited in the presence of the toxicant samples. In this study, the inhibition rate of bacteria for the exposure to a tested metal was calculated by the following equation:(1)$$\mathrm{Luminescence}\;\mathrm{inhibition}\;(\%)=\frac{\left(L_{o}-L_{i}\right)}{L_{o}}\times 100$$
where L_o_ represents luminescence (RLU) of the control sample, and L_i_ defines the luminescence (RLU) of the sample exposed to metal or mixtures of metals.

#### 2.5.3. Mathematical Modeling for Assessment of Combined Heavy Metals Toxicity

The combined effect of two or three metals eventuates into three types of joint action. They were designated into additive effect, synergistic effect and antagonistic effect [38,39]. In this study, the joint action of each combined metals was determined by Toxicity Unit and Mixture Toxic Index approaches [40,41,42].

##### Toxicity Unit (TU)

In this method, emerging ICs of the individual compounds are determined. Toxic Unit (TU) can be defined as the incipient IC_50_ for each compound. Toxic Unit (TU) of each chemical in a mixture is calculated by Equations (2) and (3), respectively
(2)$${\mathrm{TU}}_{i}=\frac{C_{i}}{{\mathrm{IC}}_{50i}}$$
(3)$$M=\sum_{i=1}^{n}{\mathrm{TU}}_{i}=(\frac{C_{1}}{{\mathrm{IC}}_{50\;1}}+\frac{C_{2}}{{\mathrm{IC}}_{50\;2}}\ldots +\frac{C_{n}}{{\mathrm{IC}}_{50\;n}})$$
whereby C_i_ represents the concentration of ith compound when the mixture is at its IC_50_, and IC_50_i is the concentration that elicits median inhibition concentration when the compound is individually tested. TU_i_ is the toxic unit of ith component in the mixture. In Equation (3), M is the sum of the toxic units TU_i_. The effect of joint toxicity calculated by M value is characterized in Table 1.

##### Mixture Toxic Index (MTI)

The other approach used to assess the joint toxicity effect was Mixture Toxic Index (MTI), which can be calculated by the Equations (4) and (5)
(4)$$M_{o}=\frac{M}{{\mathrm{TU}}_{i}\max}$$
(5)$$M_{o}=\frac{\left(1-\log M_{o}\right)}{M_{o}}\;$$

In Equation (4), M_0_ represents the ratio of M and max (TU_i_), the maximum value of toxic units TU_i_ of the single components in the mixture. Table 2 summarizes the values calculated by using the MTI approach and the resultant interactions of the compound in the mixture [33,42].

## 3. Results

### 3.1. Acute Toxicity of Combined Heavy Metals Toxicity

Table 3 show IC_50_s over acute and chronic toxicity of six metals against the bacteria before carry out mixture metals toxicity in this study. Figure 2a, Figure 3a, Figure 4a, Figure 5a, Figure 6a, Figure 7a and Figure 8a, Figure 9a, Figure 10a, Figure 11a, Figure 12a, Figure 13a, Figure 14a show the Dose-Inhibition response curve of both individual and mixture metals studied in the acute test respectively constructed by nonlinear regression models in GraphPad Prism 7.04 in terms of IC_50_s. TU (Toxicity Unit) and MTI (Mixture Toxicity Index) approaches were used for predicting the mixture toxicity between two and three metals in this study. As documented in (TU) concept, toxic units of a single metal tested in this study, and the sum of toxic units in a mixture (M) were computed as listed in (Table 4) in acute test. Afterward, data of M and MTI were compared as proven in (Table 5) and similar action was predicted by the two models.

The results suggested that the acute effect combination of Hg^2+^ + Ag^+^, Hg^2+^ + Cu^2+^, Ag^+^ + Cu^2+^, Hg^2+^ + Ag^+^ + Cu^2+^ were antagonistic, as IC_50_s are higher than the TU values as M > 1 and MTI < 1 (Table 1&2). The sigmoidal curve in (Figure 8a) indicates the inhibition increase at mixed concentrations of Hg^2+^ and Ag^+^ checked from 0.0816 mg/L until 189 16.73 mg/L.

In particular, as was the case with individual metals, the inhibition was less abrupt on the impact of binary mixture compared with individual metal Hg^2+^ and Ag^+^ tested at high concentration (Figure 2a and Figure 3a). In the same way, the antagonistic effect of mixture metal tested Hg^2+^ and Cu^2+^ on the bacteria was observed with increased concentrations from 0.327 mg/L until a high concentration 15.67 mg/L was reached (Figure 9a). Meanwhile, (Figure 10a) illustrates the dose-inhibition response curve acute toxicity of metal mixture Ag^+^ and Cu^2+^.The responses produced a sigmoid curve as inhibition increase with mixed concentration from 0.326 mg/L up to 10.12 mg/L. Likewise, acute toxicity of a tertiary mixture of Hg^2+^, Ag^+^ and Cu^2+^ on the bacteria was observed to increase the luminescence inhibition from 0.489 mg/L to 18.61 mg/L (Figure 11a). Generally, the inhibition of those metals in the tertiary mixture was less gradual when metals are combined together at high concentration. Above all, as it was the case with individual metals associating with three high toxicity metals Hg^2+^ , Ag^+^ and Cu^2+^ on the bacteria tested in this study as demonstrated in Table 3, the inhibition was less abrupt on the impact in studying mixture metal rather than individually tested metal.

Acute exposure to Cu^2+^ and Zn^2+^ mixture showed an additive effect as the sum of toxicity units, M, showed a value equal to 1 and MTI equal to 1. The influence of acute toxicity of the joint metal Cu^2+^ and Zn^2+^ produced a dose-inhibition response curve presented in (Figure 12a). The inhibition of metal Zn^2+^ in the binary mixture at low concentration was more gradual when combined with Cu^2+^ if compared to individually tested Zn^2+^. Similarly, antagonism was observed for Cd^2+^ + Cu^2+^ in acute test on the bacteria as M > 1 and MTI < 1. Figure 13a demonstrates the inhibition rapidly occurring between 24.49 mg/L and 302.04 mg/L of both Cd^2+^ and Cu^2+^. The action of metal Cd^2+^ on the luminescence of the bacteria was more gradual at low concentration when combined with Cu^2+^ compared with Cd^2+^ alone in this study. On the other hand, assessment of joint toxicity of Ni^2+^ + Zn^2+^ demonstrated that the acute test of the metal shows antagonistic effect in both approaches, TU and MTI. The impact of Ni^2+^ and Zn^2+^ was increased concentrations of both metals at mixed concentrations from 10.21 mg/L until a high concentration of 489.79 mg/L (Figure 14a).

### 3.2. Chronic Toxicity of Combined Heavy Metals Toxicity

In the multi-component mixtures, the toxicity of any compounds might change for the extended period of time they mix with the other metals because they would interact in some way. Additional studies with longer exposure times (6 h) were carried out to predict the interaction between metals exposed to chronic action.

(Figure 2b, Figure 3b, Figure 4b, Figure 5b, Figure 6b, Figure 7b) and (Figure 8b, Figure 9b, Figure 10b, Figure 11b, Figure 12b, Figure 13b, Figure 14b) show the Dose-Inhibition response curve of both individual and mixture metals studied in the chronic test respectively. The results were used to calculate the combined metal effects over 6 h exposure on the bacteria with the approaches of TU and MTI already discussed for chronic toxicity tests (Table 6 and Table 7). Antagonistic effects were observed for Hg^2+^ + Ag^+^, Hg^2+^ + Cu^2+^, Ag^+^ + Cu^2+^, Hg^2+^ + Ag^+^ + Cu^2+^, Cu^2+^ + Zn^2+^ and Cd^2+^ + Cu^2+^ according to M value > 1 and MTI lower than 0. The sigmoidal dose-inhibition response curve shown in (Figure 8b) presents the inhibition increase with the mixed concentration of Hg^2+^ and Ag^+^ for the range of concentration checked from 0.408 mg/L to 6.12 mg/L. Furthermore, it can be seen that chronic toxicity of a mixture Hg^2+^ and Cu^2+^ in this study demonstrated an inhibitory effect that gradually increased at mixed concentration range from 0.324 mg/L to 15.67 mg/L at which a slope in (Figure 9b) reaches maximum percentage inhibition. The impact of combined metal Ag^+^ and Cu^2+^ on the bacteria under chronic toxicity produces a sigmoidal dose-inhibition response curve in (Figure 10b). The responses of the tertiary mixture of metals Hg^2+^, Ag^+^ and Cu^2+^ are shown in (Figure 11b). The mixtures deliberately inhibited the luminescence activity as the concentration increased from 0.489 mg/L and seemingly reaching saturation by a maximum inhibition at 24 mg/L. In the same way, the sigmoidal curve in (Figure 12b) indicates the impact of joint metals Cu^2+^ and Zn^2+^ under chronic toxicity. Precisely, the inhibition increased rapidly at range of mixed concentration from 1.02 mg/L to 5.10 mg/L. (Figure 13b) illustrates the dose-response relationship curve of metals Cd^2+^ and Cu^2+^ in this study. The sigmoidal curve shows 97.04% inhibition at 275.51 mg/L. In the case of interaction between Ni^2+^ + Zn^2+^, synergistic effect arose under chronic exposure according to TU value < 1 and MTI greater than 1. The toxicity assessment of metals Ni^2+^ and Zn^2+^ under chronic test produced responses in (Figure 14b). The sigmoidal curve shows complete inhibition on the bacteria as the slope falls off at flat plateau at 12.24 mg/L.

## 4. Discussion

### 4.1. Acute and Chronic Action for Individual Metals on Photobacterium sp.NAA-MIE

At present, the highest metal toxicities on the bacteria are Hg^2+^ in both acute and chronic test, and the lowest toxicities are demonstrated by metal Cd^2+^ in acute test and Zn^2+^ in chronic test (Table 3). Mercury is known as the most toxic heavy metal [43,44]. It enters biomembranes easily due to its lipophilic properties. After its discharge into the environment, inorganic mercury is incited by bacteria, thereby accelerating to form methylmercury, (MeHg) and has the ability to bioaccumulate in fish and other animal tissues [43,45]. Whereas, numerous study have reported, Gram-negative bacteria is less sensitive towards Cd^2+^. The outer layer of the bacteria membrane is made of exo-polysaccharides which are able to adsorb and trap Cadmium [46,47,48]. Data in the previous study further supported the least metal toxicities of Cd^2+^ in this study. This may prevent the interaction between Cd^2+^ and key enzymes, and luciferase, which is responsible for bioluminescence production, and ultimately decrease the toxicity of Cd towards the bacteria. In addition, based on previous literature, some metals have been proven as a detrimental element for living organisms and the main integral of metabolic enzymes, such as Zn and Cu [49,50,51]. These can be toxic to organisms when exceeding threshold levels. In this study, the bacteria might profit at low concentration when Cu^2+^ and Zn^2+^ were acutely exposed, whereas high concentrations at more than IC_50_ value of 2.943 mg/L and 80.57 mg/L for Cu and Zn respectively (Table 3) were observed to increase the percentage of luminescence inhibition.

### 4.2. Comparison of Acute and Chronic Action for Metal Mixtures on Photobacterium sp.NAA- MIE

The degree of toxicity of pollutants is pertained not just to the exposure concentration, but also to the exposure time, which is a significant influencing factor. The effect of the acute and chronic joint is different thereby toxicity data regarding the mixtures in both 30 min and 6 h exposures were compared and are given in (Table 8).

Antagonistic effect of combination of the Hg^2+^ + Ag^+^, Hg^2+^ + Cu^2+^, Ag^+^ + Cu^2+^, Hg^2+^ + Ag^+^ + Cu^2+^ arise from both acute and chronic action on luminescence. The previous section has shown that metals Hg^2+^, Ag^2+^ and Cu^2+^ were most toxic to bacteria when tested individually as listed in (Table 3). In combination toxicology, the primary way to specify the joint action of the components in the mixture is to conduct experimental studies correlating the effect of the mixture to the effect of the individual compounds. However, when those metal were mixed, a lower toxicity than expected was exhibited. In this event, the interaction effect of Hg^2+^ + Ag^+^, Hg^2+^ + Cu^2+^, Ag^+^ + Cu^2+^, Hg^2+^ + Ag^+^ + Cu^2+^ resulted in a weaker effect. These interactions may decrease the toxicity of the active compound, thus their mixture exerts a weaker effect than predicted [52]. Authors of [53] argue that antagonistic interaction occurs due to the competition for binding sites in the biological interface. In this study, the reference metals Hg^2+^, Ag^+^ and Cu^2+^ are three pollutants that exhibited high sensitivity when tested individually to the bacteria for both acute and chronic tests. It can be hypothesized that the metals Hg^2+^, Ag^+^ and Cu^2+^ respectively displayed in the mixtures directly aim at luciferase, which is important in chemical reactions for luminescence production in both acute and chronic actions. Thus resulting in the fierce competition between them, which may impede their interaction with the proteins and ultimately reduce luminescence inhibition. Another toxicity study involving antagonistic interaction of mercury with selenium suggested selenium to lower the effect and depress mercury in fish tissues [54].

Similarly, an antagonistic effect was observed for the mixture of metal Cd^2+^ and Cu^2+^ from acute to chronic action on *Photobacterium* sp. NAA-MIE as TU shows M > 1 and MTI < 0. In this study, Cu^2+^ was observed to be more toxic than Cd^2+^ when tested individually. Capability of Cd^2+^ to antagonize the toxic effects of Cu^2+^ was observed. According to [52], the possibility of a partial agonist and full agonist may occur if two substances present in the mixture in a manner of low efficacy of one substance compete with the high efficacy of second substance respectively. Antagonistic interaction similarly observed in studies made by [13,21] based on inhibition determined by Microtox ^®^ assay, under acute toxicity of Cd + Cu against *P. phosphoreum* T3S with (15 min IC_50_ 8.581 mg/L) and *V. fischeri* respectively after 15-min exposure (Table 9). However, most previous studies regarding the luminescence inhibition test have only showed the interactive effect of the combined metals from 5 min to 30 min. Limited available data reported on the chronic effect. At present the disfigurement of the conventional short-term concerning substances with a slowed impact creates interest essentiality for chronic toxicity.

In the case of toxicity action of Cu^2+^ + Zn^2+^, the toxic action changes from additive action in 30 min to antagonistic action in 6 h. The additive effect of the two combined heavy metals was due to the same action mechanism, differing only in their potencies. Evaluation of the combined heavy metals toxicity may not be similar in various studies. Other authors (Table 9) reported that acute toxicity of Cu^2+^ + Zn^2+^ on *V. fischeri* manifested synergistic interaction as 15 min EC_50_ for Zinc in presence of 0.125 mg/L of copper, and was 0.14 mg/L [55]. Surprisingly, after chronic tests, both mixtures of Cu^2+^ and Zn^2+^ showed antagonistic interaction suggesting that Cu may enhance the Zinc toxic effect after 6 h exposure as Zn showed lower sensitivity than Cu to *Photobacterium* sp.NAA-MIE. Results in the present study further supported the idea of cellular uptake of Cu^2+^ as greater than that of Zn^2+^, which affected the luciferase in the luminescence bioprocess. The majority of published articles regarding the toxicity of Cu and Zn mixtures have been reported using different organisms [33,49,55]. Taken for example, acute toxicity of Zinc pyrithione (ZnPT) and Cu against three marine organisms, based on algal growth inhibition test of *Thalassiosira pseudonana* under 96-h exposure time and mortality rate of polychaete larvae *Hydroides elegans*, and amphipod *Elasmopus rapax* under 48-h and 24-h exposure time respectively presented synergistic interaction [49] in Table 9. Evaluation of mixture toxicants may not give similar interaction for different organisms due to low reproducibility, consequently it takes a long time to give a response rather than applying luminescence inhibition tests that gives fast results. Additionally, the use of luminescence inhibition bioassay has a shorter time advantage compared to other methods. According to [58], among various species (algae, crustaceans, rotifers, bacteria and protozoan), the bacterial bioluminescence assays showed the highest sensitivity in acute toxicity measurement for most of the samples.

Furthermore, [33] in Table 9 reported that the mixture of zinc and copper on *Clarias gariepinus* under 96-h exposure time showed a different reaction, for example antagonism occurred when the metals were mixed at the ratio of 1:1 and synergism at the ratio of 1:2. This study suggested that it is important to take into consideration the effect of low and high concentrations of toxicant. At present, there are limited data on different concentration ratios of constituent metal components in a mixture which influences metal toxicity to *Photobacterium* sp.NAA-MIE, thus additional studies are required in this field.

The interaction between Ni^2+^ and Zn^2+^ metals became more complex over chronic action as synergistic effect was observed. Synergistic interactions of chemicals in mixtures boost the action of other chemicals, so that they allow to jointly strive for a stronger effect than expected [59]. Even though metal Ni^2+^ and Zn^2+^ appeared to be slightly toxic when tested individually, the information theoretically proves the fact that metals forming complexes have better penetrability with regards to the bacterial cell compared to a single metal acting solely. On that account, the eventual toxicity of the mixture brings to bear a stronger effect rather than the degree of toxicity of the single metal. Besides, the concentration for both metals tested is low in mixture compared to being tested individually. It is assumed in this study that particular concentrations of heavy metals regarded as being low and of minimal effect possibly draw out more degree of toxicity in mixture with other metals. Similarly, synergism was observed by Zn and Ni mixtures on *Pimephales promelas* under standard 96-h acute tests for a 4 times concentration of Zn and a 3 times concentration of Ni, as 67% rate of mortality of the organism was predicted [58]. Other studies did not correspond well with the trend in this work. For instance, the mixture chronic toxicity of zinc and nickel to *D. magna* reproduction toxicity test for 21-days exposure time is antagonistic at a low concentration however, it becomes synergistic at the high concentration as predicted by global statistical analysis [59]. Besides, different mixture interactions of metal studied also show that the toxicity mixture present in binary or tertiary joint relies on the tested organisms [53]. Different mixture interactions were observed especially when the metals Cu^2+^ + Zn^2+^ and Ni^2+^ + Zn^2+^ were acutely and chronically exposed in this study.The author cited that mixture effect under chronic tests was dissimilar to those from acute tests as short term exposures tend not to represent metal interactions occurring with longer-term exposures.

Even so, studies on mixture have declared that interactions might be inconsistent throughout various experiments. The interactions count on concentration, types of chemicals mixed and organisms tested [60,61]. Interactions between metals mostly affect some modes that are essential for the differential responses of test organisms to mixtures of metals towards luminescence production, such as bioavailability, internal transportation, binding at the target site and interactions of the toxic elements with luciferase complex in charge for the luminescence. Furthermore, the biological makeup of the receiving organism and physicochemical nature of the metal decide the bioavailability of the metal that will act upon in the organism [62].

An approach for threat identification and risk evaluation of a complex mixture such as in medication therapies interested in vitro experiments of many antibiotics and their combinations or appropriate drug combinations will be governed to assess a public health problem.These drug combinations have derived into additive, synergistic or antagonistic interactions [63,64]. In such a way, the study on mixture compounds is beneficial to heal, delay the progression, or decrease the symptoms of diseases. Another good example that demonstrates this trend is the information regarding toxic action by antibiotic agents on *V. fischeri* from acute synergism to chronic antagonism, which suggests that the time intermission of medication should be reduced to prevent chronic antagonism [17].

## 5. Conclusions

In the current work, the tropical luminescent bacterium *Photobacterium* sp.NAA-MIE luminescent bacteria test allows the prediction of the joint effect of metal mixture according to the simultaneous analysis of two toxicological endpoints which are acute luminescence inhibition after 30 min, and chronic luminescence inhibition after 6 h. The application of two different mathematical Toxicity Unit (TU) and Mixture Toxicity Index (MTI) observes mixture toxicity of Hg + Ag, Hg + Cu, Ag + Cu, Hg + Ag + Cu and Cd + Cu, showing an antagonistic effect over acute and chronic tests. A binary mixture of Cu + Zn was observed to show additive effect at the acute test and antagonistic effect at chronic test, while the mixture of Ni + Zn showed antagonistic effect during the acute test and synergistic effect during the chronic test. The similarities and differences in prediction of the joint effect of metal mixtures with other standard bioassay in this study indicate possible different toxicity/inhibition mechanisms for metals in different organism under acute to chronic test. In future, additional studies should be conducted to quantify the antagonistic and synergistic effect of metals in the mixture to understand the mechanism of action and potency of each compound.

However, the advantage of the bioluminescence test in this study is applicable in terms of the sensitivity of the bacteria in much shorter time than other assays, is cost-effective, and may be used to counter to the possible environmental risks. Thus, the toxicity result of this strain may be effective to take into theoretical consideration toxicity levels of mixture of pollutants that occur together in ecosystems and in the same way contribute extrapolation studies under acute to chronic exposures for mixture metal toxicity in deriving safe limits and standards aimed at protecting organisms in the environment.

## Figures and Tables

**Figure 1 ijerph-18-06644-f001:**
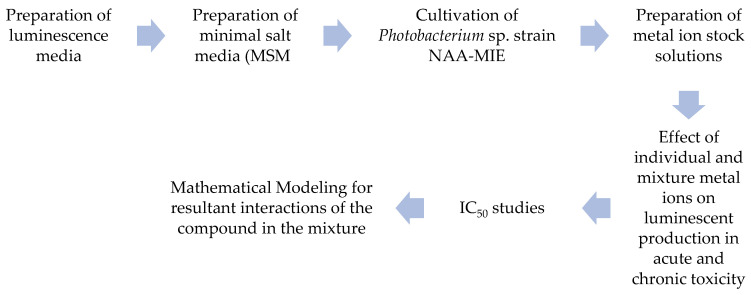
Steps in the experimental procedure for the determination of interaction existing between compounds in acute and chronic toxicity of *Photobacterium* sp.NAA-MIE against 6 heavy metal.

**Figure 2 ijerph-18-06644-f002:**
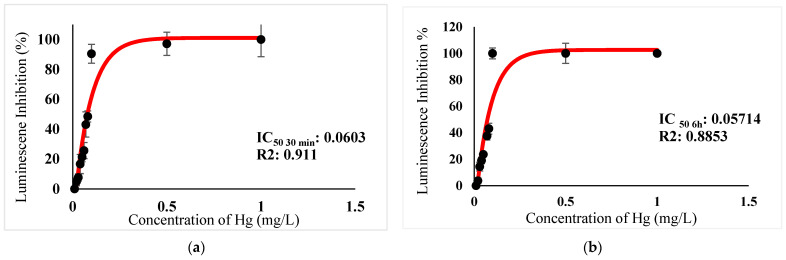
(**a**) Acute and (**b**) Chronic toxicity of Hg^2^ on luminescence inhibition of *Photobacterium* sp.NAA-MIE. Data represent mean ± SD, *n* = 3.

**Figure 3 ijerph-18-06644-f003:**
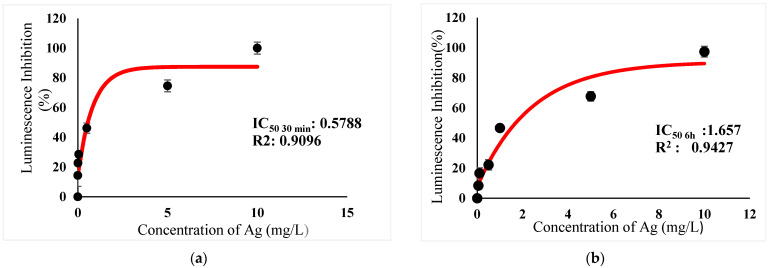
(**a**) Acute and (**b**) Chronic toxicity of Ag^2+^ on luminescence inhibition of *Photobacterium* sp.NAA-MIE. Data represent mean ± SD, *n* = 3.

**Figure 4 ijerph-18-06644-f004:**
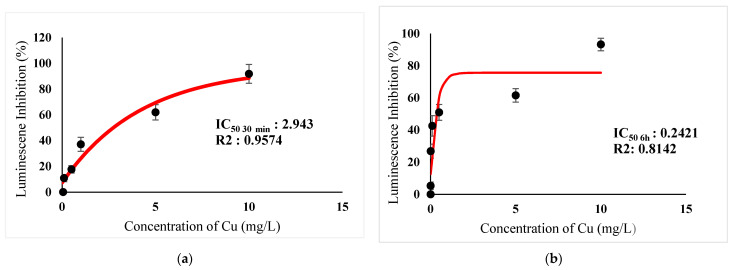
(**a**) Acute and (**b**) Chronic toxicity of Cu^2+^ on luminescence inhibition of *Photobacterium* sp.NAA-MIE. Data represent mean ± SD, *n* = 3.

**Figure 5 ijerph-18-06644-f005:**
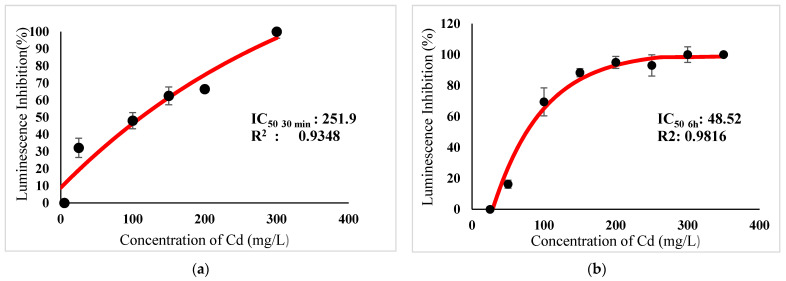
(**a**) Acute and (**b**) Chronic toxicity of Cd^2+^ on luminescence inhibition of *Photobacterium* sp.NAA-MIE. Data represent mean ± SD, *n* = 3.

**Figure 6 ijerph-18-06644-f006:**
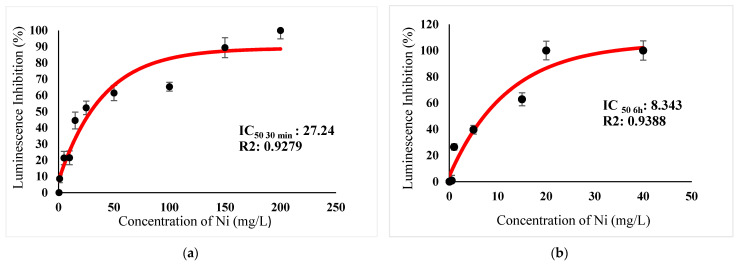
(**a**) Acute and (**b**) Chronic toxicity of Ni^2+^ on luminescence inhibition of *Photobacterium* sp.NAA-MIE. Data represent mean ± SD, *n* = 3.

**Figure 7 ijerph-18-06644-f007:**
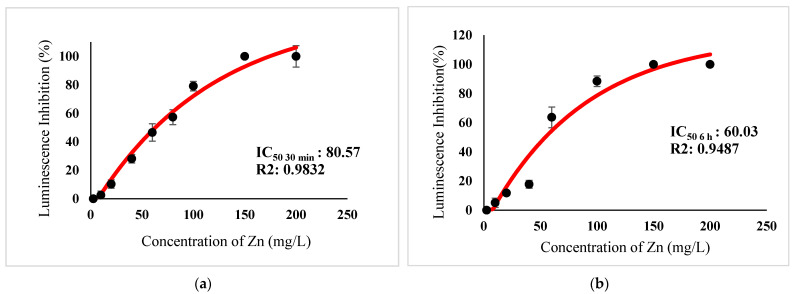
(**a**) Acute and (**b**) Chronic toxicity of Zn^2+^ on luminescence inhibition of *Photobacterium* sp.NAA-MIE. Data represent mean ± SD, *n* = 3.

**Figure 8 ijerph-18-06644-f008:**
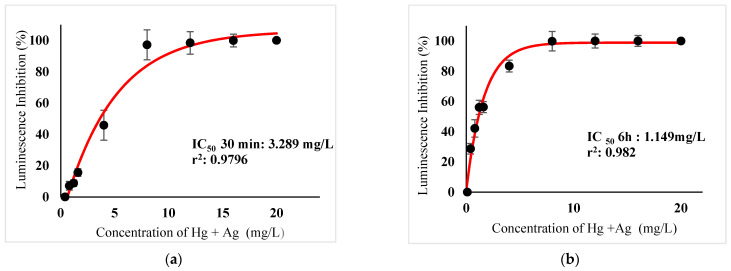
(**a**) Acute and (**b**) Chronic toxicity of mixture Hg^2+^ + Ag^+^ in concentration ratio of 1:1 on luminescence inhibition of *Photobacterium* sp.NAA-MIE. Data represent mean ± SD, *n* = 3.

**Figure 9 ijerph-18-06644-f009:**
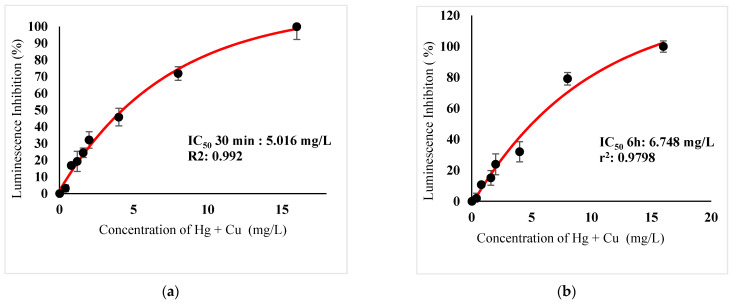
(**a**) Acute and (**b**) Chronic toxicity of mixture Hg^2+^ + Cu^2+^ in concentration ratio of 1:1 on luminescence inhibition of *Photobacterium* sp.NAA-MIE. Data represent mean ± SD, *n* = 3.

**Figure 10 ijerph-18-06644-f010:**
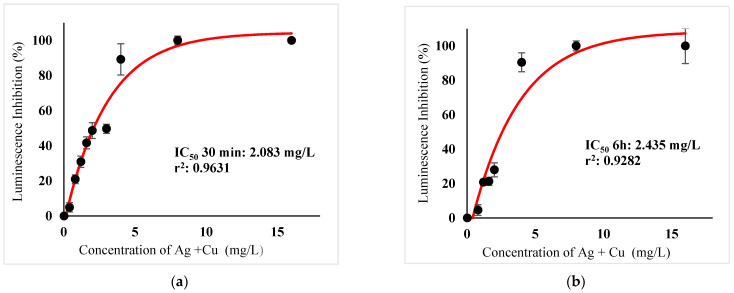
(**a**) Acute and (**b**) Chronic toxicity of mixture Ag^+^ + Cu^2+^ in the concentration ratio of 1:1 on luminescence inhibition of *Photobacterium* sp.NAA-MIE. Data represent mean ± SD, *n* = 3.

**Figure 11 ijerph-18-06644-f011:**
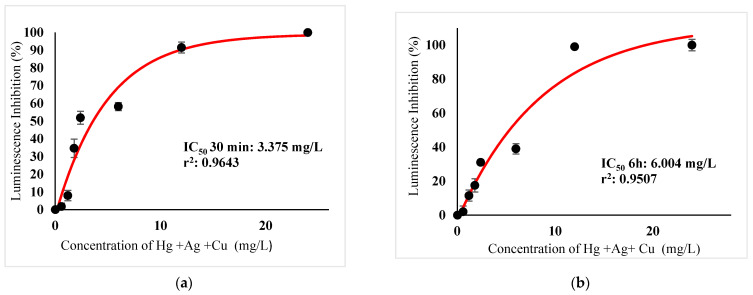
(**a**) Acute and (**b**) Chronic toxicity of mixture Hg^2+^ + Ag^+^ + Cu^2+^ in the concentration ratio of 1:1:1 on luminescence inhibition of *Photobacterium* sp.NAA-MIE. Data represent mean ± SD, *n* = 3.

**Figure 12 ijerph-18-06644-f012:**
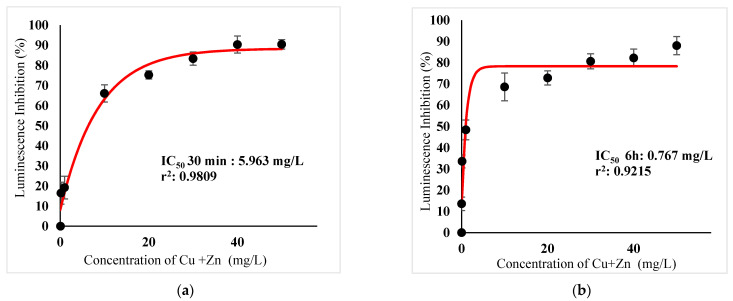
(**a**) Acute and (**b**) Chronic toxicity of mixture Cu^2+^ + Zn^2+^ in the concentration ratio of 1:1 on luminescence inhibition of *Photobacterium* sp.NAA-MIE. Data represent mean ± SD, *n* = 3.

**Figure 13 ijerph-18-06644-f013:**
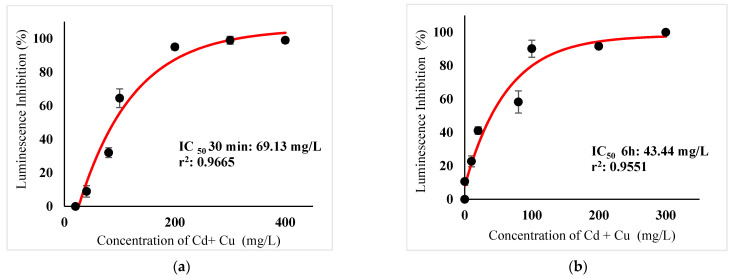
(**a**) Acute and (**b**) Chronic toxicity of mixture Cd^2+^ +Cu^2+^ in the concentration ratio of 1:1 on luminescence inhibition of *Photobacterium* sp.NAA-MIE. Data represent mean ± SD, *n* = 3.

**Figure 14 ijerph-18-06644-f014:**
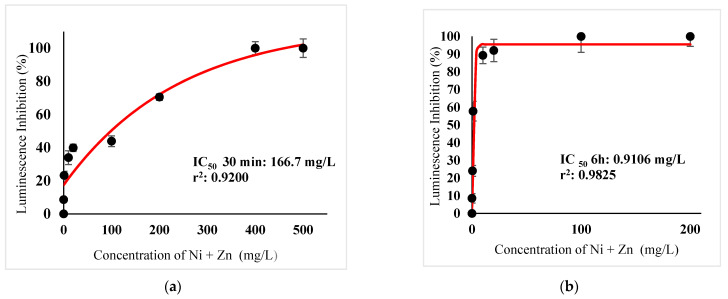
(**a**) Acute and (**b**) Chronic toxicity of mixture Ni^2+^ + Zn^2+^ in the concentration ratio of 1:1 on luminescence inhibition of *Photobacterium* sp.NAA-MIE. Data represent mean ± SD, *n* = 3.

**Table 1 ijerph-18-06644-t001:** Sum of Toxicity Unit values representing the interaction existing between mixture compound [40,41].

M (Sum of Toxic Unit)	Interaction
M < 0.8	Synergism
M between 0.8–1.2	Additive
M > 1.2	Antagonism

**Table 2 ijerph-18-06644-t002:** Mixture toxicity indices and interaction existing between compounds [42].

MTI (Mixture Toxicity Index)	Interaction
MTI < 0	Antagonistic
MTI = 0	No addition
0 < MTI < 1	Partial additive
MTI = 1	Additive
MTI > 1	Synergistic (Supra-additive)

**Table 3 ijerph-18-06644-t003:** IC_50_s over acute and chronic toxicity of six metals against *Photobacterium* sp.NAA-MIE.

Metal	*Photobacterium* sp.NAA-MIE IC 50 30 min (mg/L)	*R2*	*Photobacterium* sp.NAA-MIE IC 50 6 h (mg/L)	*R2*
Hg^2+^	0.0603	0.911	0.05714	0.8853
Ag^+^	0.5788	0.9096	1.657	0.9427
Cu^2+^	2.943	0.9574	0.2421	0.8142
Cd^2+^	251.9	0.9348	48.52	0.9816
Ni^2+^	27.24	0.9279	8.343	0.9388
Zn^2+^	80.57	0.9832	60.03	0.9487

**Table 4 ijerph-18-06644-t004:** List of Toxic Unit (TU) for each heavy metals tested acutely and a total of the toxic units in two and three joint mixtures of heavy metals on *Photobacterium* sp.NAA-MIE.

Mixture	IC_50_ (30 min)	Toxic Unit of Individual Heavy Metals						Sum of Toxic Units in a Mixture (M)
		TU _Hg_	TU _Ag_	TU _Cu_	TU _Cd_	TU _Zn_	TU _Ni_	
Hg^2+^ + Ag^+^	3.289	27.272	2.840	-	-	-	-	30.112
Hg^2+^ + Cu^2+^	5.016	41.592	-	0.8522	-	-	-	42.444
Ag^+^ + Cu^2+^	2.083	-	1.799	0.3539	-	-	-	2.153
Hg^2+^ + Ag^+^ + Cu^2+^	3.375	18.657	1.944	0.3823	-	-	-	20.983
Cu^2+^ + Zn^2+^	5.963	-	-	1.013	-	0.037	-	1.050
Cd^2+^ + Cu^2+^	69.13	-	-	11.745	0.1372	-	-	11.882
Ni^2+^ + Zn^2+^	166.7	-	-	-	-	1.035	3.060	4.095

**Table 5 ijerph-18-06644-t005:** MTI approach to determine the acute effects (30 min) of two and three combined heavy metals on *Photobacterium* sp.NAA-MIE.

Mixture	Acute Toxicity			
	M	Interactive Effect	MTI	Interactive Effect
Hg^2+^ + Ag^+^	30.112	Antagonistic	−33.4	Antagonistic
Hg^2+^ + Cu^2+^	42.444	Antagonistic	−188.25	Antagonistic
Ag^+^ + Cu^2+^	2.153	Antagonistic	−3.265	Antagonistic
Hg^2+^ + Ag^+^ + Cu^2+^	20.983	Antagonistic	−24.843	Antagonistic
Cu^2+^ + Zn^2+^	1.050	Additive	1	Additive
Cd^2+^ + Cu^2+^	11.882	Antagonistic	−206.473	Antagonistic
Ni^2+^ + Zn^2+^	4.095	Antagonistic	−3.842	Antagonistic

**Table 6 ijerph-18-06644-t006:** List of Toxic Unit (TU) for each heavy metal in chronic toxicity test and sum of the toxic units, M in mixtures of heavy metals on *Photobacterium* sp.NAA-MIE.

Mixture	IC_50_ (6 h)	Toxic Unit of Individual Heavy Metals						Sum of Toxic Units in a Mixture (M)
		TU _Hg_	TU _Ag_	TU _Cu_	TU _Cd_	TU _Zn_	TU _Ni_	
Hg^2+^ + Ag^+^	1.149	10.061	0.3467	-	-	-	-	10.408
Hg^2+^ + Cu^2+^	6.748	59.048	-	13.936	-	-	-	72.984
Ag^+^ + Cu^2+^	2.435	-	0.7351	5.031	-	-	-	5.766
Hg^2+^ + Ag^+^ + Cu^2+^	6.004	35.019	1.208	8.265	-	-	-	44.492
Cu^2+^ + Zn^2+^	0.7675	-	-	1.585	-	0.006	-	1.591
Cd^2+^ + Cu^2+^	43.44	-	-	89.715	0.447	-	-	90.162
Ni^2+^ + Zn^2+^	0.9106	-	-	-	-	0.008	0.055	0.063

**Table 7 ijerph-18-06644-t007:** TU and MTI approaches to determine the interaction of two and three combined heavy metal effects in chronic (6 h) exposure tests mixture on *Photobacterium* sp.NAA-MIE.

Mixture	Chronic Toxicity			
	M	Interactive Effect	MTI	Interactive Effect
Hg^2+^ + Ag^+^	10.408	Antagonistic	−22.666	Antagonistic
Hg^2+^ + Cu^2+^	72.984	Antagonistic	−19.429	Antagonistic
Ag^+^ + Cu^2+^	5.766	Antagonistic	−11.856	Antagonistic
Hg^2+^ + Ag^+^ + Cu^2+^	44.492	Antagonistic	−14.827	Antagonistic
Cu^2+^ + Zn^2+^	1.591	Antagonistic	−115.32	Antagonistic
Cd^2+^ + Cu^2+^	90.162	Antagonistic	−901.58	Antagonistic
Ni^2+^ + Zn^2+^	0.063	Synergistic	22.342	Synergistic

**Table 8 ijerph-18-06644-t008:** TU and MTI approaches to determine the interaction of two and three combined heavy metal effects over Acute (30 min) and Chronic (6 h) on *Photobacterium* sp.NAA-MIE.

Mixture	Acute Toxicity				Chronic Toxicity			
	M	Interactive Effect	MTI	Interactive Effect	M	Interactive Effect	MTI	Interactive Effect
Hg^2+^ + Ag^+^	30.112	Antagonistic	−33.4	Antagonistic	10.408	Antagonistic	−22.666	Antagonistic
Hg^2+^ + Cu^2+^	42.444	Antagonistic	−188.25	Antagonistic	72.984	Antagonistic	−19.429	Antagonistic
Ag^+^ + Cu^2+^	2.153	Antagonistic	−3.265	Antagonistic	5.766	Antagonistic	−11.856	Antagonistic
Hg^2+^ + Ag^+^ + Cu^2+^	20.983	Antagonistic	−24.843	Antagonistic	44.492	Antagonistic	−14.827	Antagonistic
Cu^2+^ + Zn^2+^	1.050	Additive	1	Additive	1.591	Antagonistic	−115.32	Antagonistic
Cd^2+^ + Cu^2+^	11.882	Antagonistic	−206.472	Antagonistic	90.162	Antagonistic	−901.58	Antagonistic
Ni^2+^ + Zn^2+^	4.095	Antagonistic	−3.842	Antagonistic	0.063	Synergistic	22.342	Synergistic

**Table 9 ijerph-18-06644-t009:** Comparative metals mixture toxicity studies on *Photobacterium* sp.NAA-MIE with those reported by other investigators among different species.

Mixture	30-min Acute Toxicity *Photobacterium* sp.NAA-MIE	6 h-Chronic Toxicity *Photobacterium* sp.NAA-MIE	Other Studies	
	Interactive Effect	Interactive Effect	Organism	Interactive Effect
Hg + Ag	Antagonistic	Antagonistic	Not applicable	-
Hg + Cu	Antagonistic	Antagonistic	Not applicable	-
Ag + Cu	Antagonistic	Antagonistic	Not applicable	-
Hg + Ag + Cu	Antagonistic	Antagonistic	Not applicable	-
Cu + Zn	Additive	Antagonistic	*Clariasgariepinus* [33] *Thalassiosira pseudonana, Hydroides elegan*, and *Elasmopus rapax* [49] *Vibrio fischeri* [55]	Antagonistic ^a^ Synergistic ^a^ Synergistic ^a^
Cd + Cu	Antagonistic	Antagonistic	*P. phosphoreum* [13] *Vibrio fischeri* [21]	Antagonistic ^a^ Antagonistic ^a^
Ni + Zn	Antagonistic	Synergistic	*Pimephales promelas* [56] *Daphnia magna* [57]	Synergistic ^a^ Antagonistic ^b^

^a^ acute toxicity test ^b^ chronic toxicity test.

## Data Availability

Not applicable.

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
