# Peer review of "Comparison of Joint Effect of Acute and Chronic Toxicity for Combined Assessment of Heavy Metals on Photobacterium sp.NAA-MIE"

_ijerph, 2021, doi:10.3390/ijerph18126644_

Round 1

Reviewer 1 Report

Thank you for providing the revised version which has still issues and is still difficult to rea due to missing clarity. Moreover, the references must be sorted throughout to be in ordered line in the text.

The scope of the manuscript is still unclear and must be higlighted in a comprehensive way. Why was investigated Photobacterium sp.NAA-MIE while such kind of investigations would have better been performed on bacterium that is well known? Literature references on Photobacterium sp.NAA-MIE other that one own references are missing and must be added.

Added figure 1 is not a comprehensive flow chart and doesnt help to streamline the methodology. A real flow chart should be in the beginning of the methodology section, and contain detailed information on the methods.

The required layout demands are missing in the text and the tables.

As I already recommended, I repat to separate the discussion section from the results section as the text is quite confused and confusing. I throngly recommend to follow a red line and to condense the results section.

I do not really agree the conclusions as there are no comprehensive studies for comparison and the data can not be retraced by the reader.

Author Response

refer attachment

Reviewer 2 Report

Dear authors, 

I think your experimental work it' s interesting, but the overall quality of the manuscript should be greatly improved, in particular the data analysis and presentation of the results and discussion section. In the attached file you can find specific comments and suggestions. 

Author Response

refer attachment

Round 2

Reviewer 1 Report

Thank you for providing the revised version, which still has issues. All tables must be transformed into the MDPI format. Furthermore, my comment was not considered, why the investiagtion was done with an "exotic" bacterium. Fom my point of view, this is the main reason why the manuscript is not interesting for the scientific ommunity. Except the authors, nearly nobody worked with that bacterium and nobody benefits scientifically from the results. And by the way, nobody can proof the results, because of the same reason.

Having in view the given comments, the scope of the investigation remains still unclear. Reason is also the low level literature research. The scope of the lierature research is to identify gaps in the scientific literature that require the presented work. This was not done, and it remains unclear for which purpose the investigation was done. I strongly recommend to clarify this point.

Moreover, the first paragraph under section 1 does not have any connection with the subject and must be replaced with relevant information.

Section 2.5.2: why the formula is in bold? Shall this highlighting express something?

Table 4: there is talked about the "Sum of Toxic Units In a mixture (M)". This part must be explained, and scientifically explained. This issue remains completely unclear throughout the manuscript. Ag, Cu, Ni, and Zn are essential elements, while Hg and Cd are non essential elements, and are usually considered toxic. Obviously, in the investigation, all of them are considered toxic, which might be the case also for the essential elements, but only in very high concentrations. This issue must be considered and explained in a scientifically sound way. Particularly, it must be explained what is a "toxic unit" and under which conditions the above mentioned essential elements are considered toxic. The explanation must be supported by scientific evidence.

Author Response

refer attachment

This manuscript is a resubmission of an earlier submission. The following is a list of the peer review reports and author responses from that submission.

Round 1

Reviewer 1 Report

Thank you for submitting your manuscript to the International Journal
of Environmental and Public Health. Generally, the manuscript fits into
the scope of the journal, however, the structure respect Scientific
Best Practice as the discussion is missing. The layout of the tables
does not respect MDPI format. Furthermore, there are some comments
that require revision.

In the literature review, it is important that the scientific novelty of the work is established through a critical analysis of related literature. How does this work contribute towards the gaps identified? How does it improve upon previous work? It is recommended that a short discussion of the novel contribution of each reference cited be provided to give readers a better understanding of their relevance. The scope of the manuscript must be formulated in a clear way.

The methodology must be improved. I strongly recommend to include a flow chart illustrating the steps of the methodology. The interviews must be documented.

The discussion section is missing and must be added.

The conclusions are by far too short. In the conclusions, in addition to summarising the actions taken and results, please strengthen the explanation of their significance. It is recommended to use quantitative reasoning comparing with appropriate benchmarks, especially those stemming from previous work.

Reviewer 2 Report

Dear authors,

in the attached file you can find my suggestions and comments. 

In my opinion the manuscript and the data still requires hard work; it's also important you check the quality of your data fitting.

Best regards
